# Genetic Stability Assessment of Six Cryopreserved Strawberry (*Fragaria* × *ananassa* Duch.) Accessions by Phenotypic and Molecular Studies

**DOI:** 10.3390/biology11121746

**Published:** 2022-11-30

**Authors:** Jinjoo Bae, Yunseo Choi, Jae-Young Song, Jung-Ro Lee, Munsup Yoon, Young-Yi Lee

**Affiliations:** 1National Agrobiodiversity Center, National Institute of Agricultural Sciences, RDA, Suwon 16613, Republic of Korea; 2Department of Horticultural Bioscience, College of Natural Resources and Life Science, Pusan National University, Busan 50463, Republic of Korea

**Keywords:** cryobanking, droplet vitrification, ISSR marker, long-term conservation, somaclonal variation

## Abstract

**Simple Summary:**

Cryopreservation technology has been developed, demonstrating safe, efficient, and economic benefits for the long-term preservation of genetic resources. It receives more attention because dramatic germplasm losses have been observed in field conservation, caused by many factors such as climate change, natural hazards, and diseases. Therefore, demands for cryobanking crops have been increasing. Cryopreservation methods have been developed for strawberries, the most preferred berry fruit, due to their nutritional and economic values. However, during the cryopreservation process, the explants are exposed to many steps under stressful conditions, which may induce somaclonal variations. The article describes a genetic stability assessment by the greenhouse performance of vegetative and fruit traits and molecular marker analysis for six strawberry accessions. Our results suggest that the cryopreservation protocol used in this study can be applied to implement cryobanking of strawberry germplasm.

**Abstract:**

For the long-term preservation of genetic resources, cryopreservation techniques have been developed for strawberry germplasm, mainly using in vitro-grown shoot tips. In this study, genetic stability was tested under greenhouse conditions for six strawberry accessions (IT232511, PHS0132, IT245810, IT245830, IT245852, and IT245860) derived from the following procedures: (1) conventional propagation (GH: greenhouse maintained); (2) in vitro propagation (TC: tissue culture); (3) pretreatment before cryopreservation (−LN: non-liquid nitrogen exposure); and (4) cryopreservation (+LN: liquid nitrogen exposure). To test the performance of phenotypic traits, we measured six vegetative and five fruit traits. There were no distinct differences in most of the characteristics, but a few traits, such as sugar content and pH of fruits in three accessions, showed higher values in +LN compared to GH. However, the differences disappeared in the first runner generation. To test genetic variations, a total of 102 bands were generated by twelve inter simple sequence repeat (ISSR) primers. A few polymorphic bands were found only in plants derived from TC of IT245860, which was not cryopreserved. The sequencing analysis of four polymorphic bands produced by ISSR_15 showed that none of these sequences matched the characterized genes in NCBI. Phenotypic abnormality was not observed across all plants. This study indicates that cryopreserved plants of the six strawberry accessions are phenotypically and genetically stable. Therefore, the results of this study can help to implement cryobanking of strawberry germplasm.

## 1. Introduction

Strawberry (*Fragaria* × *ananassa* Duch.) is one of the most important crops in many countries because of its nutritional and economic benefits. Reflecting such importance, the production, consumption, and development of new cultivars continue to grow across the world. The conservation of germplasm has received increasing attention for maintaining the genetic diversity of ecology and for securing genetic resources to develop a new cultivar with agronomic and economic value. Field genebank and in vitro preservations are commonly used for strawberries. However, field preservation risks germplasm loss due to contamination, climate disasters, and diseases. It also demands intensive labor and cost for maintaining a large field. Although in vitro preservation can overcome these issues, the primary concern of the method is the occurrence of somaclonal variations. Studies have reported that somaclonal incidences in strawberries can be influenced by exogenous plant growth regulators (such as 6-benzylaminopurine, cytokinins), the number of subcultures, explants, and genotypes [1,2,3,4,5,6]. Bairu et al. [7] have identified somaclonal variations of in vitro propagated plants in approximately 100 plant species, indicating serious concerns when using this method.

Recent progress in cryopreservation technology has demonstrated the safe, efficient, and economical advantages of long-term preservation. Therefore, the technology has undergone significant development during the last two decades, enabling the cryobanking of many crops such as apple, apricot, banana, blueberry, and garlic [8,9,10,11]. The cryopreservation protocols for strawberry have been developed using different methods [12,13,14,15,16]. However, several studies observed off-types or subtle genetic variability of cryopreserved strawberries that were affected by high sucrose concentrations or toxic plant vitrification solution (PVS) 2 during cryopreservation procedures [17,18,19]. Since PVS2 contains toxic substances such as dimethyl sulphoxide (DMSO) and polyethylene glycol (EG), PVS3 consisting of glycerol and sucrose, was developed and is widely used for plant materials that are sensitive to chemical toxicity [20]. To implement cryobanking with developed protocols, it is necessary to evaluate the genomic integrity of plants derived from cryopreservation [21].

Assessments of genetic integrity have been performed at various levels, including phenotypic, histological, cytological, biochemical, molecular, and genetic levels [22,23,24]. Morphological and biometric assessments have been widely used to evaluate the true to-type and thus easily detect agronomic characteristics such as flower color, leaf morphology, and pigmentation abnormality, plant stature, and yield [7,23]. To date, only a few studies have considered the morphological performance of cryopreserved strawberry plants [19,25].

A molecular-level assessment can detect more subtle changes in the DNA that would not be observed in a phenotype [26]. Random DNA markers such as amplified fragment length polymorphism (AFLP), inter simple sequence repeat (ISSR), and random amplified polymorphic DNA (RAPD) have been widely used for studies of genetic stability [18]. Among these markers, ISSRs have been indicated to be more reliable for strawberry cultivars because the nature of the high sugar and polyphenol content of strawberry may limit the reliable use of other markers [26]. Due to the rapid development of these techniques, molecular markers have been broadly used to assess genetic stability [27,28,29,30]. Therefore, phenotypic approaches with supportive evidence from molecular analysis have been conducted to assess stability [1,7,31].

Our previous study [12] reported that droplet vitrification methods could be effectively applied to thirty-one accessions of in vitro-grown strawberry shoot tips. The present study was carried out to investigate the genetic stability of six randomly chosen strawberry accessions comprised of a wild relative, a developed line, and four developed cultivars. Plants were derived from three major procedures during cryopreservation, which are TC (tissue culture; in vitro propagated and maintained by subculture), −LN (nonliquid nitrogen exposure; pretreatment before cryopreservation), and +LN (liquid nitrogen exposure; cryopreservation). Each of these procedures plays a crucial role in the incidence of somaclonal variations. These plants were then compared to GH (conventionally propagated plants in a greenhouse). Since few studies have evaluated greenhouse strawberry plants regrown from cryopreservation, the objective of this study was to evaluate the genetic stability of greenhouse strawberry plants regrown from cryopreservation using morphological characteristics (six vegetative and five fruit traits) and 12 ISSR markers.

## 2. Materials and Methods

### 2.1. Preparation of Plant Material

Strawberry accessions (*Fragaria* × *ananassa* Duch.) were introduced into the greenhouse in the National Agrobiodiversity Center in Suwon (longitude 127°0′32′′ E, latitude 37°17′28′′ N), Republic of Korea. Six accessions (IT232511, PHS0132, IT245810, IT245830, IT245852, and IT245860) that looked healthy and maintained a sufficient number of plants after cryopreservation were randomly chosen for this study (Table 1).

Mother plants were maintained in the greenhouse. Runner tips of 1–2 cm were excised from mother plants during the growing season in the greenhouse. Collected runner tips were disinfected by immersing in 70% ethanol for 3 min and then in 1.0% (*v*/*v*) sodium hypochlorite (NaOCl) for 10 min. They were then rinsed three times with sterile distilled water. In vitro plants were initiated from 1–2 mm long shoot tips excised from disinfected runner tips. Explants were cultured on shoot multiplication medium composed of Murashige and Skoog (MS) medium [32] supplemented with 0.2 mg·L^−1^ 6-benzylaminopurine (BAP, Duchefa Biochemie, Haarlem, The Netherlands), 30 g·L^−1^ sucrose, and 8.0 g·L^−1^ plant agar (Duchefa Biochemie, Haarlem, The Netherlands). The pH of the medium was adjusted to 5.8 before autoclaving at 121 °C for 15 min. Subcultures were performed every 6 weeks. All cultures were incubated at 24 ± 1 °C under a 16 h photoperiod with a light intensity of 50 μmol·m^−2^·s^−1^ provided by cool-white, fluorescent lamps. After multiplication, shoots with uniform sizes were selected and moved onto hormone-free MS medium supplemented with 30 g·L^−1^ sucrose and 2.6 g·L^−1^ phytagel (Sigma-Aldrich, Saint Louis, MO, USA) adjusted to pH 5.8 (hereafter ‘standard medium’). After 6 weeks of culture, plants were then used for cryopreservation experiments.

### 2.2. Cryopreservation Using Droplet Vitrification

Cryopreservation procedures were described in detail in our previous study [12]. Shoot tips (1–2 mm in length) were excised from a six-week-old in vitro grown plant in a standard medium. The explants were precultured in liquid MS medium containing 0.3 M sucrose for 31 h followed by preculturing in liquid MS medium containing 0.5 M sucrose for 17 h. Precultured explants were treated with 35% PVS3 (17.5% of glycerol + 17.5% of sucrose) as loading solution for 40 min and then 100% PVS3 (50% of glycerol + 50% of sucrose) dehydration solution for 1 h [20]. A treated explant was placed in each droplet of 2.5 µL PVS3 placed on a sterilized aluminum foil strip and immediately plunged in LN for at least 1 h. After LN immersion, aluminum foil with shoot tips was transferred into 0.8 M sucrose unloading solution (40 °C) for 40 min. Rewarmed shoot tips were cultured on regrowth medium composed of NH_4_NO_3_-free MS medium supplemented with 1.0 g·L^−1^ casein, 1.0 mg·L^−1^ GA_3_, 0.5 mg·L^−1^ BAP, 30 g·L^−1^ sucrose, and 2.6 g·L^−1^ phytagel (pH 5.8) for five days in the dark followed by transferring to normal MS medium supplemented with 1.0 g·L^−1^ casein, 0.5 mg·L^−1^ GA_3_, 30 g·L^−1^ sucrose and 2.6 g·L^−1^ phytagel (pH 5.8) for 9 days under low light intensity for recovery. Surviving shoot tips were transferred to standard medium. +LN were obtained from regrown shoots after cryopreservation. −LN plants were obtained from regrown shoots after presolution treatments without liquid nitrogen immersion. TC plants were developed from the excised shoot tips directly cultured on a standard medium.

### 2.3. Acclimatization of Regrown Plants and Evaluation of Vegetative Characteristics

After regrown plants derived from TC, −LN, +LN were rooted, eight-week-old in vitro-grown plants were removed from containers, washed off the media with running water, and then moved to a greenhouse (Figure 1a). These plants were planted into 72-cell trays filled with peat-based soil and acclimatized with a plastic cover for 3 weeks under a relative humidity of more than 80% (Figure 1c). The cover was then removed. When plants developed new roots, they were transferred into plastic pots (8.5 cm (D) × 8 cm (H)) containing peat-based soil (Figure 1d). In order to compare these plants with conventionally propagated plants, newly developed young plants in the greenhouse were detached from mother plants (GH) (Figure 1b). Plantlets with sizes comparable to those of plants from others (TC, −LN, +LN) were planted into pots at the same time. The experiment was performed using a completely randomized block design containing five plants with three replications of GH, TC, −LN, and +LN. Plants grew in the pots for three months, after which six vegetative traits, including plant length, leaf length, leaf width, petiole length, the number of leaves, and the number of runners, were evaluated on 27 August 2021 (Figure 1e). The most developed leaf was chosen from each plant and was measured for leaf length, width, and petiole length.

### 2.4. Evaluation of Fruit Characteristics and Quality Traits under Greenhouse Conditions

After vegetative characteristics were measured, three individual plants from each replication were transplanted into containers (18 cm (W) × 57 cm (L) × 15 cm (H)) (Figure 1f). During the summer period (July–August), plants produced the runners. Three runners from plants in each replication were collected and developed into plants. When runners were rooted, plants were transplanted into containers having the same size. Thus, two groups of plants grew during the rest of the year in 2021 and produced flowers from January to May of the following year. We harvested fruits during March and April 2022, when plants produced the most fruits (Figure 1g). Fruit characteristics were evaluated by measuring fresh weight, width, length, total sugar content, and pH. Fruits were harvested from originally developed experiments in addition to plants developed from runners. Total sugar contents and pH were determined using a Digital Refractometer PAL-1 (Atago Co., Tokyo, Japan) and a pH Spear (Eutech, Singapore), respectively.

### 2.5. Data Analysis

Strawberry data collection of growth and developmental traits were conducted according to instructions from the Strawberry Agricultural Research and Extension Service in Chungcheong-do, Republic of Korea. Data were collected from five plants with three replications for vegetative characteristics. Fruit characteristics were evaluated by measuring individual fruits harvested from three plants with three replications. Data were analyzed using the least significant difference (LSD) at *p* < 0.05 and Duncan’s multiple range test with SAS 7.1 software (SAS Institute Inc., Cary, NC, USA). The results of all numerical values are presented as the mean values.

### 2.6. Detection of Genetic Variation Using ISSR

Total genomic DNA was isolated from young leaves of four treatments using a GenEx™ Plant kit (GeneAll Biotechnology Co., Ltd., Seoul, Republic of Korea). Each treatment contained three DNA samples for six accessions. A total of 15 primer sets and polymerase chain reaction (PCR) conditions for amplification were given in previous studies [26,28,33] (Table 2).

Twelve (ISSR_1, ISSR_2, ISSR_3, ISSR_7, ISSR_8, ISSR_9, ISSR_10, ISSR_11, ISSR_12, ISSR_13, ISSR_14, ISSR_15) of 15 ISSR markers were used for PCR analysis because of reproducibility of their bands. PCR mixtures (20 μL) consisted of 2 μL of 10 ng·μL^−1^ genomic DNA, 13.4 μL of ddH_2_O, 2 μL of 10X e-TAQ Reaction Buffer containing 25 mM MgCl_2_ (SolGent, Daejeon, Republic of Korea), 2 μL of 10 pmol·μL^−1^ primer, 0.4 μL of each 10 mM dNTP Mix (SolGent), and 0.2 μL of Solg™ e-Taq DNA polymerase (SolGent). PCR amplification was conducted using a Veriti™ 96-Well Thermal Cycler (Thermo Fisher Scientific, Waltham, MA, USA). Reaction products were analyzed via electrophoresis on a 2% (*w*/*v*) agarose gel at a constant voltage of 150 V for 2 h. The agarose gel was stained with ethidium bromide and photographed using a Gel Image Analysis System (CoreBio-MAXTM, Davinch-K, Seoul, Republic of Korea) under UV light exposure. A 100 bp DNA ladder was used as a molecular-weight size marker.

Polymorphic bands were scored visually as presence (1) or absence (0) of each marker. Genetic similarity between accessions was estimated using Nei’s genetic distance [34]. Cluster analysis was performed by applying the unweighted pair-group method with an arithmetic mean (UPGMA) using the *Poppr* R package [35] with 100 bootstrap replicates.

### 2.7. Sequencing Analysis

PCR for cloning was carried out on a Veriti™ 96-Well Thermal Cycler (Thermo Fisher Scientific, Waltham, MA, USA) under the following conditions: denaturation at 94 °C for 5 min followed by 45 cycles of denaturation at 94 °C for 1 min, annealing at 55 °C for 1 min, extension at 72 °C for 1 min, and a final extension at 72 °C for 2 min. PCR mixtures were prepared in a total of 50 μL containing 5 μL genomic DNA (10 ng·μL^−1^), 33.5 μL ddH₂O, 5 μL 10X buffer Solg™ (SolGent, Daejeon, Republic of Korea), 5 μL ISSR_15 primer (10 pmol·μL^−1^), 1 μL dNTPs (10 mM, SolGent), and 0.5 μL e-Taq Solg™ Taq polymerase (5 U·μL^−1^, SolGent). PCR products were electrophoresed on a 2% (*w*/*v*) agarose gel at 130 V for 3 h and purified using the Expin™ Gel SV gel extraction kit (GeneAll, Seoul, Republic of Korea). Purified amplicons were cloned into a TA vector (Dyne PCR Cloning Kit, DyneBio Inc., Seoul, Republic of Korea) and transformed into *Escherichia coli* (DH5α). The plasmid DNA was isolated using an Exprep™ Plasmid SV mini kit (GeneAll, Seoul, Republic of Korea) and then sequenced using the dye-terminator method by Genotech (Daejeon, Republic of Korea). Homologous genes were identified using BLASTn of the National Center for Biotechnology Information (NCBI, www.ncbi.nlm.nih.gov, accessed on 9 October 2022).

## 3. Results

### 3.1. Greenhouse Performance

Most in vitro grown plants derived from TC (tissue culture; in vitro propagated and maintained by subculture), −LN (nonliquid nitrogen exposure; pretreatment before cryopreservation), and +LN (liquid nitrogen exposure) survived during acclimatization. Although plants were derived from different procedures, in vitro-grown plants of TC, −LN, and +LN could provide uniform plants by subculturing at the same time. It was difficult to directly compare the plants derived from tissue cultured and those grown in a greenhouse since they were developed in different environmental conditions. It was also challenging to make uniform plantlets for GH when they were detached from the mother plants. GH plants started to grow slowly and a few plants died in the early stage. There were no differences in plant length, leaf length, leaf width, and petiole length among the four treatments (Table 3).

However, in vitro-grown plants developed significantly greater numbers of leaves and runners compared to GH plants except for IT245860. For IT245852, GH plants were smaller than other treatments for all six traits due to slow growth in the early stage. However, abnormal plants were not observed. Although most GH plants of IT245852 were generally smaller, a few of them grew out that they were observably similar to others (Figure 2). Overall, the plants derived from TC, −LN, and +LN did not show differences in vegetative growth of the six accessions. In vegetative growth, differences were mainly found between GH and in vitro-grown plants (TC, −LN, +LN) for traits of the number of leaves and runners.

Most of the plants produced white flowers between February and April 2022. IT232511, a wild relative, started flowering the earliest and continued to be floriferous. It produced pink flowers with a smaller foliage size and a compact habit, which were distinct from other accessions (Figure 2). This accession did not set fruits due to self-incompatibility. Based on our observations, there were no abnormal plants or significant differences in flowering habits such as flowering time and intensity. PHS0132 set very poorly, with a small size and an oval-shaped fruit. As shown in Table 4, there was no significant difference in fruit weight, width, length, or sugar content across treatments. However, fruit pH values of +LN and –LN were higher than those of GH and TC plants. For IT245810 and IT245830, none of the traits showed significant differences across treatments (Table 4). IT245852 produced many flowers and fruits of a conical shape. Plants from TC, −LN, and +LN were different from GH plants in fruit length and sugar content (Table 4). IT245860 grew normally in vegetative growth but set poorly and produced many malformations. Fresh weight, width, and length were not significantly different across treatments for this accession. However, GH was different from the other treatments in terms of sugar content and pH (Table 4). Overall, the traits of fruit weight, fruit length, and width were stable in all six accessions. When we replicated the evaluation for plants derived from runners, we did not find differences in fruit characteristics for any of the six accessions across treatments (data not shown). Thus, the results showed that the differences produced in direct comparison among treatments were not maintained in the clonal progeny generation.

### 3.2. DNA Analysis Using ISSR Marker

An ISSR marker analysis was performed on 72 DNA samples that contained three DNA samples for each treatment of six accessions. DNAs from two samples, a GH sample of IT232511 and a +LN sample of PHS0132, were omitted because of their poor quality. The poor quality likely resulted from our sampling of somewhat more aged leaves instead of young ones. Twelve of fifteen primers successfully produced 102 discernable bands. The three primers (ISSR_4, ISSR_5, ISSR_6) failed to produce discernable bands for these strawberry accessions. Polymorphic bands were generated among six accessions that were distinguishable from each other (Figure 3). The results also showed no polymorphism between treatments and replications within an accession of five. However, five primers (ISSR_7, ISSR_10, ISSR_12, ISSR_13, ISSR_15) produced a few polymorphic bands in the TC plants for the IT245860 accession (Figure 3).

A UPGMA dendrogram (Figure 4) was prepared to show genetic distance for six accessions, generating two clusters. According to the dendrogram, there were no genetic distances within treatment or replications for five accessions. For IT245860, the Nei’s genetic distance between variants from TC and others was less than 0.1, which showed these plants were genetically similar to each other.

The inheritability of this somaclonal variation was tested. A PCR primer of ISSR_15 was selected for an extended test of IT245860 since it produced a clear polymorphic band of approximately 300 bp. Samples comprised 15 individuals derived from TC plants, 15 from cryopreserved plants, nine from runners of TC plants, and nine from runners of cryopreserved plants. The results showed no additional variants in TC or cryopreserved plants. Polymorphic bands from original TC plants were maintained in plants derived from runners of variants (data not shown).

To gain further insight into these polymorphic bands, we conducted a sequencing analysis. For sequencing analysis associated with polymorphisms, bands of approximately 300 bp, 500 bp, 650 bp, and 950 bp produced by primer ISSR_15 were selected (Figure 5). Sequencing of the 300 bp band (ISSR_15-300) and BLASTn in NCBI indicated that the sequence was 96% identical to a partially uncharacterized gene of *Fragaria vesca subsp. Vesca*, LOC101293334 in GenBank (GeneID: 101293334, NCBI) (Table 5 and Figure 6), whereas no significant matches were observed for other band sequences.

## 4. Discussion

Most cryopreservation methods of vegetatively propagated plants require three major procedures of TC, −LN, and +LN. Each of these procedures plays a crucial role in the incidence of somaclonal variations. The cryopreservation protocol used in this study attempted to minimize the factors that induced somaclonal variations. Following the recommendations from previous studies, low concentrations of BAP (0.2 mg·L^−1^) during in vitro proliferation and PVS3, instead of PVS2, for presolution treatment in cryoprocedures were applied.

The greenhouse performance of plants derived from TC, −LN, and +LN was compared with that of GH from mother plants grown in a greenhouse, and the stability of vegetative growth and fruit characteristics were evaluated. Several factors such as different culture conditions, plant age or developmental stage, and mother plant conditions of GH can affect the direct comparison between GH and in vitro grown plants. Indeed, many studies showed the different performances between GH and TC, reporting that in vitro plants produced uniform plants with superior performance compared to conventionally propagated plants [36,37,38]. In our study, in vitro-grown plants of TC, −LN, and +LN showed increased leaf numbers and runners in most accessions. Our results are consistent with many strawberry studies that reported increased numbers of leaves and runners in different strawberry cultivars of TC derived plants compared to conventionally propagated plants [39,40,41]. Waithaka et al. [42] suggested that cytokinin in the shoot proliferation medium might affect the increased runner production of in vitro grown plants due to enhanced axillary bud activity. Naing et al. [4] also reported cytokinin effects on cell division, leading to a higher number of shoots, leaves, and roots. Inferring from these studies, the increased number of leaves and runners in vitro grown plants during vegetative growth appears to be related to 0.2 mg·L^−1^ BAP application in the culture medium during in vitro proliferation of shoots. We did not find any negative effects of TC, −LN, and +LN treatments during vegetative growth.

In terms of fruit evaluation, the traits of mean weight and size of a single fruit exhibited no differences across the treatments in six accessions. The results are consistent with other studies that reported insignificant differences between in vitro cultured plants and conventionally propagated plants for these traits in different strawberry cultivars ‘Veestar’, ‘Redcoat’ [41], ‘Filon’, ‘Teressa’ [38], ‘Santa’, ‘Fanta’, ‘Berrystar’, ‘Honeybell’ and ‘Okhyang’ [4]. Thus, these traits seemed stable across genotypes. In three accessions, significant differences were found in fruit quality traits in the first growing cycle. Total sugar content and pH are two essential factors related to the fruit taste, which represents fruit quality. The pH of PHS0132 and the sugar content of IT245852 showed higher values in +LN and −LN than in GH and TC. Thus, presolution treatments for cryopreservation were affected. On the other hand, in vitro propagation affected pH and sugar contents in IT245860. Our results indicated that different accessions were affected at different levels by treatments. Our findings are related to the strawberry study of Media et al. [25], who evaluated fruit quality and fruit production in the field after cryopreservation. Their study reported genotype-specific responses in which two different cultivars were affected by different factors. However, in our study, those differences found in the first growing cycle disappeared when plants developed from runners were compared, suggesting that these differences were not maintained in the plants. This result coincides with other studies on cryopreserved sugarcane [43] and banana [44] in that the variation in the first culture cycle did not occur in subsequent somatic generations. It appears that different culture conditions and environmental stress during cryopreservation affected the plants at different levels, especially in the early stage, but these differences were not retained in the plants. These results thus suggest that the plants derived from four procedures during cryopreservation were phenotypically stable.

Molecular analyses can provide supportive evidence for genetic stability. They eliminate environmental effects and detect subtle changes in the DNA level. ISSRs have been successfully applied to genetic stability studies in strawberry [28,45] as well as in many other crops such as Lilium [29], melon [27], pineapple [30], sugarcane [46], and tomato [33]. Our result that observed a few polymorphic bands only in the TC plants of IT245860 indicates that this particular accession was sensitive to certain factors during in vitro proliferation conditions. Many strawberry studies reported the induction of somaclonal variations during TC, mostly with higher than 0.2 mg·L^−1^ BA or BAP applications [1,2,3,4,5,6]. Unlike in these studies, somaclonal variation was generated with low BAP concentrations (0.2 mg·L^−1^) in our study. Interestingly, polymorphic bands were observed only in TC but not in −LN or +LN employed for in vitro culture before or after LN treatments. These results might be due to the small sample size of this study. Otherwise, variated plants might not survive when they were regrown after −LN and +LN treatments. Further research needs to address this assumption. Based on these results, cryopreserved strawberries of six accessions were considered genetically stable by molecular assessment using ISSR markers, similar to many molecular studies of cryopreserved plants in various crops [27,29,30,46].

According to the sequencing analysis using BLASTn in NCBI, only one out of four polymorphic bands was matched to an uncharacterized gene of *Fragaria vesca*. These results indicate that the polymorphic bands might have been derived from noncoding regions that would not affect the phenotype. Our phenotypic evaluation further supported that there was no significant variation among treatments. Several studies reported similar findings of somaclonal variations by DNA markers but no phenotypic changes in date palms [47], rice [48], and strawberry [49]. Kaity et al. [50] mentioned that the current molecular techniques cover only a small portion of the genome, and hence, a detected variation is not representative of the entire genome. They also mentioned that polymorphic bands could potentially be caused by transposable elements or point mutation. In addition, detected variations may lie in non-coding regions and may not affect phenotypic characteristics and development.

## 5. Conclusions

The present study showed genetic stability with phenotypic and ISSR marker analysis of six strawberry germplasms recovered from cryopreservation. The greenhouse performance evaluation for phenotypic characteristics can be informative for these results because greenhouse cultivation is the most widely implemented for strawberry production in the Republic of Korea. Our phenotypic evaluation suggests that at least the first clonal progeny generation is necessary to verify whether changes are maintained. ISSR markers detected a genetic distance of fewer than 0.1 variations in the TC treatment of IT245860. This variation did not affect phenotypes. The assessment methods we used in this study, greenhouse performance of phenotypic traits and DNA marker analysis, offered supporting evidence for our conclusion that the cryopreserved strawberry accessions were phenotypically and genetically stable. These results indicated that the cryopreservation protocol effectively worked with 0.2 mg·L^−1^ BAP for in vitro propagation, PVS3 vitrification solution for pretreatment, and droplet-vitrification methods for LN treatment. The results of this study could help to implement cryobanking of strawberry germplasm. In the future, the stability of these plants needs to be studied over more than one clonal generation if it is maintained or not. It would also broaden our understanding with applications to other assessments of cytological or biochemical levels.

## Figures and Tables

**Figure 1 biology-11-01746-f001:**
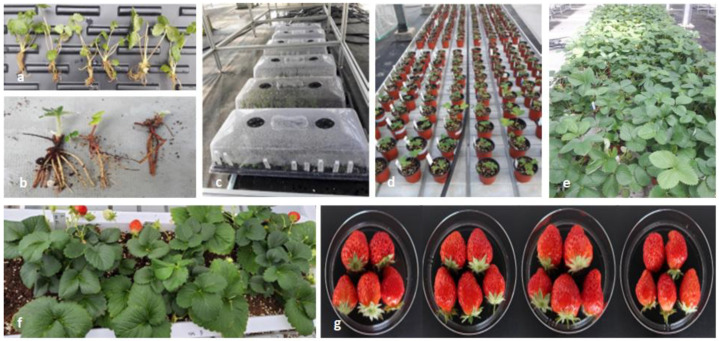
Strawberry plants grown in the greenhouse: (**a**) In vitro grown plants before acclimatization; (**b**) Plants detached from the mother plants (GH); (**c**) Acclimatization; (**d**) Early stage of vegetative growth; (**e**) At the stage of vegetative growth evaluation; (**f**) The beginning of the fruit set; (**g**) Harvested fruit of IT245810 from the four treatments (orders from the left: GH, greenhouse maintained; TC, tissue culture; +LN, liquid nitrogen exposure; −LN, non-liquid nitrogen exposure).

**Figure 2 biology-11-01746-f002:**
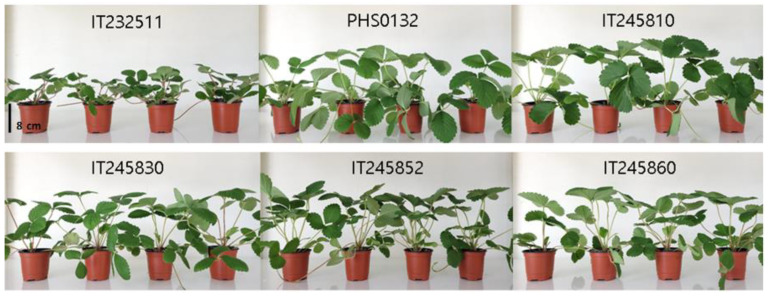
Vegetative growth comparisons of four treatments for six strawberry accessions (plant orders from the left in each accession; GH, greenhouse maintained; TC, tissue culture; +LN, liquid nitrogen exposure; −LN, non-liquid nitrogen exposure). Photos were taken three months after the planting in the pot.

**Figure 3 biology-11-01746-f003:**
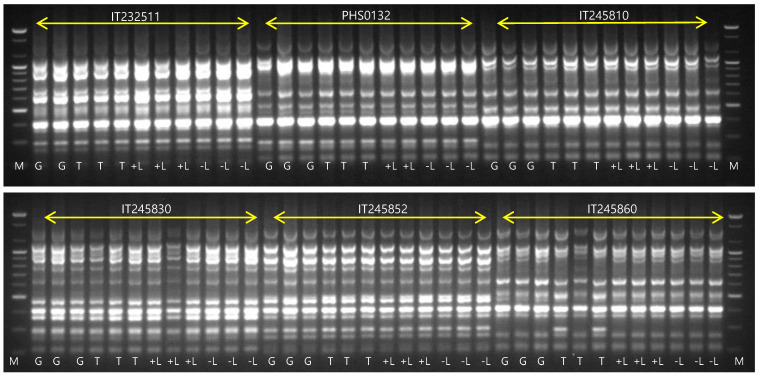
Comparisons of PCR band patterns of ISSR 15 primers obtained from six strawberry accessions from four different treatments (M, 100 bp size marker; G, greenhouse maintained; T, tissue culture; +L, liquid nitrogen exposure; −L, non-liquid nitrogen exposure). Each treatment contained three DNA samples, but two samples were missing due to poor DNA qualities (a GH sample of IT232511 and a +LN sample of PHS0132).

**Figure 4 biology-11-01746-f004:**
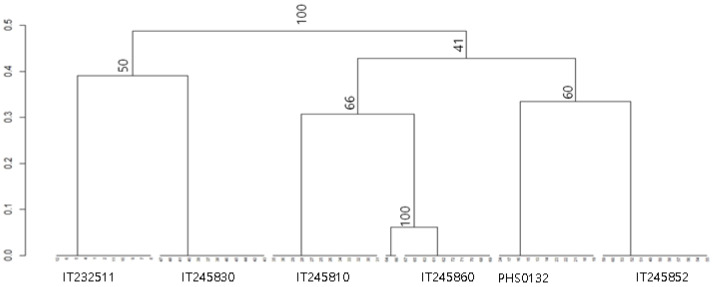
A UPGMA dendrogram showing the genetic distance of six strawberry accessions containing three samples of four treatments (GH, greenhouse maintained; TC, tissue culture; +LN, liquid nitrogen exposure; −LN, non-liquid nitrogen exposure). * The bootstrap percentage values are shown on nodes.

**Figure 5 biology-11-01746-f005:**
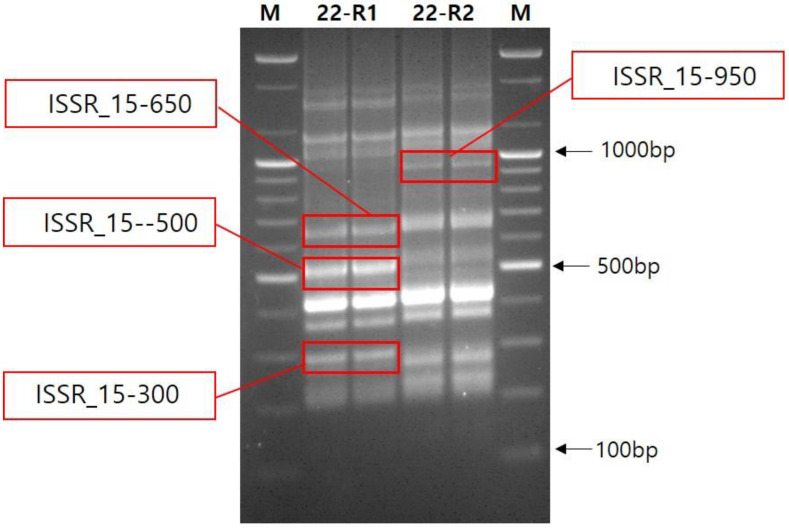
An agarose gel run for sequencing of four polymorphic bands (in red rectangles) produced by ISSR_15 primer from IT245860 accession with TC treatment. M, 100 bp size marker. 22-R1: variant, 22-R2: standard.

**Figure 6 biology-11-01746-f006:**
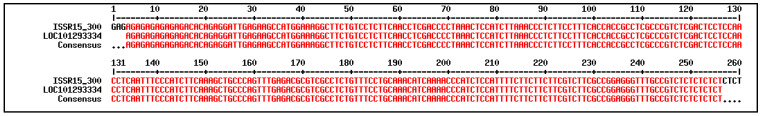
The sequence alignment of ISSR_15_300 polymorphic band produced with a partial uncharacterized gene of *Fragaria vesca* subsp. *vesca*, LOC101293334, by using Multalin software (http://http://multalin.toulouse.inra.fr/multalin/ assessed on 25 May 2022) for comparison.

**Table 1 biology-11-01746-t001:** The six strawberry accessions used in this study.

Genebank No.	Name	Status	Origin
IT232511	Pink paend	Wild relatives	USA
PHS0132	Gorella	Developed Varieties	USA
IT245810	NY1406	Developed Line	USA
IT245830	Merrimack	Developed Varieties	unknown
IT245852	Tangi	Developed Varieties	unknown
IT245860	Tufts	Developed Varieties	unknown

**Table 2 biology-11-01746-t002:** List of ISSR markers used in this study for screening the six strawberry accessions.

No	Marker	Primer Sequence ^a^	Tm (°C)	References
1	ISSR_1	VBV(AC)_7_	51	[26]
2	ISSR_2	BDB(CA)_7_	51	[26]
3	ISSR_3	HBH(CT)_7_	47	[26]
4	ISSR_4	GCV(TC)_7_	49	[26]
5	ISSR_5	BDV(AG)_7_	47	[26]
6	ISSR_6	(AG)_8_C	45	[28]
7	ISSR_7	(GA)_8_T	45	[28]
8	ISSR_8	(CA)_8_A	45	[28]
9	ISSR_9	(AC)_8_G	45	[28]
10	ISSR_10	(AG)_8_YA	45	[28]
11	ISSR_11	(GATA)_2_(GACA)_2_	45	[28]
12	ISSR_12	(AC)_8_YG	55	[33]
13	ISSR_13	(AG)_8_YT	55	[33]
14	ISSR_14	(CA)_8_RC	55	[33]
15	ISSR_15	(GA)_8_C	55	[33]

^a^ V = non T, B = non A, H = non g, D = non C.

**Table 3 biology-11-01746-t003:** Effects of four treatments on six vegetative traits for six strawberry accessions.

Accession	Treatment ^z^	Plant Length ^y^ (cm)	Leaf Length (cm)	Leaf Width (cm)	Petiole Length (cm)	No. of Leaves	No. of Runners
IT232511	GH	12.00 ^NS^	4.60 ^NS^	4.00 ^NS^	6.63 ^NS^	7.25 b	4.25 b
	TC	12.10	4.37	4.16	6.17	10.20 a	7.67 a
	+LN	12.03	4.43	4.27	6.10	9.80 a	7.73 a
	−LN	11.86	4.18	3.98	6.00	9.36 a	7.27 a
PHS0132	GH	21.38 ^NS^	7.50 ^NS^	6.12 ^NS^	12.38 ^NS^	6.25 b	3.00 b
	TC	21.53	7.68	6.19	11.89	7.43 a	4.43 a
	+LN	20.17	7.67	6.10	10.83	7.67 a	5.67 a
	−LN	21.91	7.91	6.53	12.37	7.93 a	5.40 a
IT245810	GH	21.20 ^NS^	6.44 b	6.40 ^NS^	12.70 ^NS^	6.00 b	2.80 b
	TC	21.20	6.95 a	6.75	12.27	7.60 a	4.47 a
	+LN	20.25	7.10 a	6.57	11.80	7.73 a	5.20 a
	−LN	20.25	7.38 a	6.25	12.13	7.50 a	4.75 a
IT245830	GH	21.50 ^NS^	7.40 ^NS^	5.10 ^NS^	12.10 ^NS^	6.20 b	3.20 b
	TC	20.56	7.24	4.84	11.66	8.36 a	5.00 a
	+LN	20.92	7.42	5.10	11.17	8.27 a	4.80 a
	−LN	20.43	7.53	5.13	11.10	9.00 a	4.30 a
IT245852	GH	14.17 b	3.82 b	2.90 b	9.42 b	5.00 b	2.83 b
	TC	20.19 a	6.35 a	4.96 a	12.23 a	8.00 a	5.08 a
	+LN	19.93 a	6.75 a	5.17 a	11.85 a	8.00 a	5.14 a
	−LN	20.31 a	6.29 a	4.88 a	12.00 a	8.62 a	5.00 a
IT245860	GH	21.04 ^NS^	6.58 ^NS^	6.10 ^NS^	12.94 ^NS^	7.20 ^NS^	5.80 ^NS^
	TC	21.75	6.34	5.37	13.47	7.67	5.93
	+LN	20.50	6.32	5.43	12.68	7.93	6.07
	−LN	21.98	6.23	5.51	13.50	8.00	6.33

^z^ GH, greenhouse maintained; TC, tissue culture; +LN, liquid nitrogen exposure; −LN, non-liquid nitrogen exposure. ^y^ Data presented as mean. Different letters within a column denote significantly different according to Duncan’s multiple range test (*p* < 0.05); ^NS^ Not significant.

**Table 4 biology-11-01746-t004:** Effects of four treatments on five fruit characters for five strawberry accessions.

Accession	Treatment ^z^	Fruit Fresh Weight (g) ^y^	Fruit Width (mm)	Fruit Length (mm)	Sugar Content (Brix)	pH
PHS0132	GH	2.28 ^NS^	17.08 ^NS^	22.21 ^NS^	8.54 ^NS^	3.28 b
	TC	2.09	15.66	22.58	8.90	3.25 b
	+LN	2.13	16.23	22.30	9.07	3.48 a
	−LN	2.10	15.74	21.63	8.69	3.40 a
IT245810	GH	7.31 ^NS^	24.69 ^NS^	29.91 ^NS^	7.28 ^NS^	3.82 ^NS^
	TC	7.90	25.37	29.80	7.36	3.78
	+LN	7.27	24.16	29.57	7.15	3.78
	−LN	6.12	23.39	29.38	7.56	3.88
IT245830	GH	3.36 ^NS^	20.37 ^NS^	19.66 ^NS^	7.10 ^NS^	3.35 ^NS^
	TC	3.82	20.53	21.53	7.11	3.26
	+LN	3.46	19.90	20.84	7.41	3.31
	−LN	3.51	20.59	20.87	7.15	3.31
IT245852	GH	5.93 ^NS^	22.33 ^NS^	30.82 a	6.22 c	3.42 ^NS^
	TC	5.41	21.70	27.68 b	6.68 bc	3.37
	+LN	5.56	21.81	28.45 b	8.28 a	3.35
	−LN	5.57	21.89	27.89 b	7.21 b	3.37
IT245860	GH	5.11 ^NS^	22.98 ^NS^	21.80 ^NS^	6.62 c	3.50 b
	TC	4.70	22.55	21.84	7.82 b	3.62 a
	+LN	4.33	21.53	20.83	8.76 ab	3.71 a
	−LN	4.25	22.12	21.91	9.02 a	3.67 a

^z^ GH, greenhouse maintained; TC, tissue culture; +LN, liquid nitrogen exposure; −LN, non-liquid nitrogen exposure. ^y^ Data presented as mean. Different letters within a column denote significantly different according to Duncan’s multiple range test (*p* < 0.05); ^NS^ Not significant.

**Table 5 biology-11-01746-t005:** The sequencing analysis of polymorphic PCR bands produced in TC treatment of IT245860 by primer ISSR_15.

No	PCR Band	Blastn ^z^			
		Organism	Type	Query Cover (%)	Organism
1	ISSR15_300	*Fragaria vesca subsp. vesca*	mRNA	96	XM_004287017.2
2	ISSR15_500	-	-	-	-
3	ISSR15_650	*Gossypium hirsutum*	gDNA	4	CP023744.1
4	ISSR15-950	-	-	-	-

^z^ Blastn in NCBI. - not matched in Blastn of NCBI.

## Data Availability

Not applicable.

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
