# Peer review of "Genetic Stability Assessment of Six Cryopreserved Strawberry (Fragaria × ananassa Duch.) Accessions by Phenotypic and Molecular Studies"

_biology, 2022, doi:10.3390/biology11121746_

Round 1

Reviewer 1 Report

Dear Authors, 

I have an opportunity to review manuscript entiled: “Genetic stability assessment of six cryopreserved strawberry (Fragaria x ananassa Duch.) accessions by phenotypic and molecular studies” submitted to the Biology MDPI journal.

Authors concentrated on long-term preservation of genetic resources for strawberry germplasm using in vitro grown through cryopreservation techniques. Authors tested genetic stability in greenhouse condition of six strawberry accessions by procedures:

-grrenhouse propagation;

-in vitro tissues culture propagation;

- pretreatment before cryopreservation;

- cryopreservation under liquid nitrogen;

Moreover, there were no observed distinct differences in most of traits but sugar content and pH of fruits in three accessions showed higher values in +LN compared to GH. However, the differences were disappeared in the first runner generation. Furthermore, sequencing analysis of four polymorphic bands produced by ISSR-15 showed that none of these sequences matched the characterized genes in NCBI. Phenotypic abnormality was not observed across all plants. Cryopreserved plants of the six strawberry accessions are phenotypically and genetically stable.

In my opinion, that it is more methodology paper rather for horticultural studies than research data for biological science

Please, precise more deeply the aim of the study;

The methods are quite good and in a repetitive way described, taking into account Author’s previous research;

The results are mainly clear but a few tables presenting results should be reformulate:

Table 3 is a quite difficult to analyses, it seems like raw data- “effect of four repeated”; Please use average values ; the same situation is with table 4;

Line 256 - Moreover, was there any differences in flowering time or intensity dependent on the used methods?

What was the effect of propagation on pH, please explain for biologist, why it is important factor in this analyses?

Sincerely

Author Response

Dear Reviewer,

We thank you for your time and comments. We tried our best to address your comments.

We understand that our study focused on more methodology than biological science but we believe that this study are relevant to the special issue Improvement and Innovation of Cryopreservation and In Vitro Methods in Plant Resources Protection.

Table 3 and 4 were revised by using only mean values with letters, which indicate significant differences.

Line 256 was updated.

As for pH: Sugar content and pH (acidity) are two essential factors in determining fruit taste. Therefore, those two traits are represent for fruit quality traits (line#362~364). Two characteristics respond to environmental effects. We wanted to see if the plants derived from the different conditions (GH, TC, -LN, +LN) can affect fruit pH and whether the effects were permanent or temporary.

We hope these changes followed your comments.

Sincerely,

Jinjoo Bae

National Agrobiodiversity Center, National Institute of Agricultural Sciences, RDA

Suwon, 16613, Republic of Korea

Reviewer 2 Report

Since the MS is well articulated and written a new methodology and I think it should be accepted it in its current form for publication in "Biology". As a whole I was unable to find any logical weakness in this paper.

Author Response

Dear Reviewer,

We thank you for your time and comment.

We carefully checked the writing  and revised the manuscript.

We appreciate your acceptance.

Sincerely,

Jinjoo Bae

National Agrobiodiversity Center, National Institute of Agricultural Sciences, RDA

Suwon, 16613, Republic of Korea

Reviewer 3 Report

Cryoconservation of strawberry germplasm is potentially very useful to reduce costs and labour needed for for strawberry germplasm conservation. The approach used in this study is not particularly novel and the results are expected, but anyway useful to confirm that this technique can be safely used.

Author Response

(The authors gave the same response as above.)

Reviewer 4 Report

The manuscript is quite interesting. It is focused on the broadly understood impact of various steps of a cryopreservation protocol on the phenotypic and genetic stability of six strawberry accessions.

The English form is generally understandable, although some grammar, style, and punctuation errors require improvement. Please, notice that MDPI uses a serial coma.

Some unclear parts of the Abstract require better explanation (see the corrected manuscript).

Keywords should be arranged alphabetically.

Some older references require updating, especially in the Introduction section. The authors are also incorrectly using some terms, such as 'variety' (should be 'cultivar') or 'sterilization' (should be disinfection'). I suggest referring to fruit crops

In the Materials and methods section, some additional information is needed. For example, producers of critical chemicals and equipment used. The unit style is incorrect.

Some parts of the Results should be transferred to the Discussion section or deleted. Moreover, occasionally, the authors should be more precise when describing the results.

I suggest adding suggestions for future research in the Conclusions.

For more specific comments, please see the corrected manuscript.

After incorporating all the changes, the manuscript can be accepted for publication.

Author Response

Dear reviewer,

We thank you for your time and constructive comments on the manuscript. We appreciate your detail guidance. We tried our best to address your comments.

  1. The English form is generally understandable, although some grammar, style, and punctuation errors require improvement. Please, notice that MDPI uses a serial coma. :
  • Corrected Fragaria x ananassa Duch → Fragaria × ananassa Duch (line #3, 42, 101)
  • Serial coma correction: line #57
  1. Some unclear parts of the Abstract require better explanation (see the corrected manuscript).
  • Line #32 (A few polymorphic bands were found only in plants derived from TC of IT245860.) : TC plants were derived from in vitro proliferation. These plants were not used for cryopreservation.
  1. Keywords should be arranged alphabetically:
  • Keywords were rearranged in alphabetic order (line #38 – 39)
  1. Some older references require updating, especially in the Introduction section. The authors are also incorrectly using some terms, such as 'variety' (should be 'cultivar') or 'sterilization' (should be disinfection'). I suggest referring to fruit crops

Updated references

  • #9 for fruit crops : Hassan, S.; Bhat, K.M.; Jan, A.; Mehraj, S.; Wani, S.A.; Khanday, M.U.D.; Bisati, I.A. Managing genetic resources in temperate fruit crops. Economic Affairs. 2018, 63, 987–996.

  • #10 for fruit crops : Kaviani, B.; Kulus, D. Cryopreservation of endangered ornamental plants and fruit crops from tropical and subtropical regions. Biology 2022, 11, 847. https://doi.org/10.3390/biology11060847

  • #24 Augusto, R.C.; Kulus, D.; Souza, A.V.; Kaviani, B.; Vicente, E.F. Cryopreservation of agronomic plant germplasm using vitrification-based methods: An overview of selected case studies. J. Mol. Sci. 2021, 22, 6157. https://doi.org/10.3390/ijms22116157

  • Variety → cultivar (line #46, 81, 90)
  • Hormones → plant growth regulators (line #52-53)
  • Sterilized → disinfected (line #109, 113)
  1. In the Materials and methods section, some additional information is needed. For example, producers of critical chemicals and equipment used. The unit style is incorrect.
  • The unit style was corrected (line #115, 116, 120, 122, 137, 139, 200, 219-221, 334, 353, 387, 388, 420)
  • Additional information was filled (line #116, 117, 122, 174)
  • Line#123 : cultures → culture
  1. Some parts of the Results should be transferred to the Discussion section or deleted. Moreover, occasionally, the authors should be more precise when describing the results.
  • Line #250 – 255; revised
  • Line# 264 was deleted
  • Line#272-275 were deleted
  • Line#289-290: ISSR_4, ISSR_5, ISSR_6 three markers did not work for these strawberry accessions so they were removed from this study.
  1. I suggest adding suggestions for future research in the Conclusions: added #422-425
  • In the future, the stability of these plants needs to be studied over more than one clonal generations if it is maintained or not. It would be also broaden our understanding by applications to other assessments of cytological or biochemical levels.  

Again, thank you for your guidance to improve our manuscript.

Sincerely,

Jinjoo Bae

National Agrobiodiversity Center, National Institute of Agricultural Sciences, RDA

Suwon, 16613, Republic of Korea
